# Study on the Development of a Conceptual Framework to Identify the Risk Factors of Diabetic Retinopathy among Diabetic Patients: A Concept Paper

**DOI:** 10.3390/ijerph191912426

**Published:** 2022-09-29

**Authors:** Nurul Athirah Naserrudin, Mohammad Saffree Jeffree, Nirmal Kaur, Syed Sharizman Syed Abdul Rahim, Mohd Yusof Ibrahim

**Affiliations:** 1Department of Community Health, Faculty of Medicine, Universiti Kebangsaan Malaysia, Jalan Yaacob Latiff, Bandar Tun Razak, Cheras, Kuala Lumpur 56000, Malaysia; 2Sabah State Health Department, Malaysia Ministry of Health, Tingkat 3, Rumah Persekutuan, Jalan Mat Salleh, Peti Surat 11290, Kota Kinabalu 88590, Sabah, Malaysia; 3Department of Public Health Medicine, Faculty of Medicine and Health Science, Universiti Malaysia Sabah, Jalan UMS, Kota Kinabalu 88400, Sabah, Malaysia

**Keywords:** diabetic retinopathy, diabetes mellitus, Sabah, risk factors

## Abstract

The most common complication of diabetes mellitus (DM) is diabetic retinopathy (DR). The control of DR risk factors is essential for the effective prevention of DR. There is currently a lack of research to guide DR-related research in Malaysia. This concept paper aimed to review published studies and propose a conceptual framework (CF) as a guide for future research to determine the prevalence of DR and its risk factors across DM patients. After a review of prior research, this study has presented a CF that takes into account these four key elements: the patient’s sociodemographic characteristics, comorbidities, complications, and diabetes conditions, namely, the length of the disease and glycaemic control. In the suggested CF, ethnicity was highlighted as a crucial risk factor for DR across lower- to middle-income countries (LMIC) and multiracial countries. In order to guide future studies, scientific guidance is essential. The proposed CF would help future research to conduct scientific research related to DR. Also, the proposed CF was tailored to suit research across LMIC and multiracial countries.

## 1. Introduction

Diabetes mellitus (DM) is one of the most significant non-communicable diseases (NCD) worldwide. Worldwide, there were more than 20 million diabetic patients in 2017 [1]. Poor management and untreated DM are linked to microvascular and macrovascular problems that can gradually harm a diabetic patient’s immune system and organs [2,3]. The common complications of DM include nephropathy, retinopathy, diabetic ulcers, and cardiovascular diseases [4]. Diabetic retinopathy (DR) is the most prevalent among these problems in diabetic patients. As many as 30.0% of diabetic patients are affected by DR, and in many nations, DR is the main factor contributing to vision impairment [5]. According to data from the World Health Organization (WHO), 10.0% of diabetic patients with a 15-years disease duration will acquire significant visual impairment, and 2.0% will go blind [6].

According to the Malaysian Diabetic Eye Registry, the prevalence of DR was 36.8%. However, these data were published by the Ministry of Health (MOH) in 2007. The recent prevalence of DR at the national level is still inconclusive. This indicates that studies on DR in Malaysia are still lacking. The paucity of scientific evidence on DR is concerning because patients might not be aware of the serious effects that DR has on their health. The lack of scientific guidance to carry out the research in a local setting is one tenable explanation for the paucity of DR research. Hence, more research on DR is needed to induce awareness of DR among patients and to guide the healthcare practitioner in instituting programmes or policies aimed at preventing DR.

In order to close the gap, this study foresees the need to produce a concept paper that would provide scientific guidance in conducting DR-related studies. A concept paper or protocol study was widely accepted as it could become guidance for future researchers to plan their studies in detail [7,8,9,10,11,12]. Therefore, this study aims to develop a conceptual framework that could be used in future research to determine the prevalence of DR and its risk factors across DM patients, specifically among lower-to middle-income countries and across multiple racial countries. The development of the conceptual framework would be according to the findings of the literature review of previous studies on DR.

This current manuscript is arranged as follows: in the following section, the researchers explain the findings of the literature reviews. Then, the researchers present the research gap, suggested conceptual framework, proposed study framework, and conclusions.

## 2. Literature Review

The literature review aimed to identify studies that reported evidence related to the background information and statistics of DR and DR risk factors. The included studies were reviewed and analysed to guide this current study in developing the conceptual framework of DR-related studies. This study did not perform a formal systematic literature review, but a systematic approach to analysing the literature was undertaken based on the Preferred Reporting Items for Systematic Review (PRISMA) guidelines. The search strategy used the terms “diabetes”, “type 2 diabetes mellitus”, “diabetic retinopathy”, and “risk factors of diabetic retinopathy”. The search was performed using Boolean operators such as “OR” and “AND”.

During the review, it was identified that most of the articles related to diabetic retinopathy were predominantly published in the early 1990s. Therefore, the literature search was limited to a timeline from the year 2000 to the present. After the literature search through the electronic databases, namely, PubMed and ScienceDirect, 56 articles were reviewed thoroughly. After further review of the inclusion and exclusion criteria (Table 1), finally, 44 articles were identified to be pertinent to the study’s objective.

The next sub-section explained the findings of the literature review. The findings were arranged according to the following theme:Background information on DR;Statistics and blindness related to DR;The risk factors of DR.

### 2.1. Background Information on DR

DR is a complication that afflicts the blood vessels of the eye. The high blood sugar level among diabetic patients may disrupt the retinal vascular circulation and negatively affect visual acuity [6]. At the early stage of the complication, DR may not show any severe symptoms. However, it can cause blindness if left untreated [13]. Commonly, blindness will ensue when vitreous haemorrhage and retinal detachment occur in the blood vessels of the eye. DR can be categorised based on its severity level, starting with clinically significant macular oedema (CSME), followed by diabetic macular oedema (DME), non-proliferative DR (NPDR), and proliferative DR (PDR). According to a past study, the majority of diabetic patients with DR were diagnosed as having non-proliferative DR [14]. In addition, among these four categories, PDR carries the most substantial risk of visual loss in patients. In addition to sudden vision loss, other permanent complications such as tractional retinal detachment and neovascular glaucoma can also occur among DM patients with PDR.

Different patients may experience different types of DR symptoms [15,16]. One of the commonly reported symptoms is fluctuating vision. Some patients also complained of having eye floaters or frequently seeing spots. Additionally, some patients experience shadows in the visual field. In addition, blurring of vision and distorted view are also among the common symptoms of DR. Moreover, patients may experience corneal abrasion. This condition can cause damage to the eye tissues and delay the healing of the damaged area, subsequently causing corneal abnormalities. Apart from that, some patients also encounter double vision and eye pain. Finally, cataracts are also possible among diabetic patients with retinopathy.

### 2.2. Prevalence of DR

The global prevalence of DR in 2021 was reported to be 22.7% [17]. Based on the in-depth analysis of the different categories of diabetic-related eye complications in the study, there was a higher prevalence of proliferative DR compared to diabetic macular oedema (50.0% versus 25.0%, respectively) [18]. From the observation of past studies, the recent prevalence of DR ranges from 14.7% to 60.2%. In 2016, the prevalence of DR among adults in the United States was 14.7% [19]. On the other hand, an Indian study reported the prevalence of DR as high as 60.2% [20]. A Pakistan study reported DR to be present in 17.5% of DM patients [21]. In China, the prevalence of DR was reported to be 18.2% [22]. In addition, a recent Singaporean study reported the prevalence of DR to be 28.2%, which showed a slight decrement from the older study, which reported the prevalence of 35.0% [23,24]. In Malaysia, the prevalence of DR was reported to be 33.5% [25]. The prevalence of DR in Spain was reported to be 14.9% and 22.5% in New Zealand [26,27]. A prevalence rate of 16.0% was reported in Saudi Arabia [28].

### 2.3. The Risk Factor

Based on the literature review, the existing studies indicated that DR is associated with the patient’s sociodemographic characteristics, clinical condition, diabetes complications, and diabetic condition.

Sociodemographic characteristics such as age, gender, and ethnicity play a crucial role in determining the diagnosis of DR. The age of the patient is among the significant factors related to DR. However, it remains inconclusive as to which age group of patients has the highest risk of developing DR. A study conducted in the United Kingdom reported a strong association between DR and the age of the patients [29]. This United Kingdom study also reported that patients above 58 years old were twice as likely to be diagnosed with DR compared to those younger than 58 years old. This finding is echoed by another study conducted in India, in which a higher prevalence of DR was detected among patients over 50 years old [30]. A Malaysian study reported the same finding and found that DR was more common among diabetic patients 62 years old and above [31]. Although the majority of the studies indicated a higher percentage of DR among older patients, the Wisconsin Epidemiological Study reported the opposite, in which the severity of DR was related to a younger age at diagnosis. In addition, a Singaporean study also reported that patients 65 years old and younger were two times more likely to be diagnosed with DR [24].

In addition to age, gender is also a significant factor associated with DR. The earlier published studies have reported a higher percentage of DR among male patients [14,29,32]. For instance, a study conducted in India highlighted that men were 1.5 times more likely to be diagnosed with DR than women [32]. A Malaysian study also reported a higher prevalence of DR among men compared to women (21.3% versus 14.6%, respectively) [14]. A similar pattern was also observed in the recently published studies, whereby the diagnosis of DR was more prevalent among men compared to females [33,34]. For instance, an Italian study has reported the prevalence of DR among men to be 14.0% higher than in women [33]. Although most of the published studies exhibited a higher prevalence of DR among men, there was also a study that reported a contradicting finding [35]. This contradicting finding could be due to the age of the study population, whereby the mean age of the samples was 61.9 [35]. The majority of the female patients in the previous study were at their postmenopausal stage, which increases the risk of DR due to the lack of oestrogen.

Another risk factor of DR that requires attention is the ethnicity of the patient. However, there is a lack of evidence in terms of the association between ethnicity and DR. Most of the studies outside Asia do not consider ethnicity as an important risk factor for DR [36,37,38,39,40,41]. Findings from the literature review have revealed that studies that explored the association between ethnicity and DR were mostly conducted across Asian countries. However, despite several published studies from the region, the findings remain inconclusive. A Malaysian study reported a higher percentage of DR among Chinese than Indians and Malays [31]. In this Malaysian study, the prevalence of DR among Chinese, Malays, and Indians was 35.5%, 33.2% and 30.4%, respectively. However, this finding contradicts another Malaysian study, where it was confirmed that the prevalence of DR is higher among Malays compared to other ethnicities [42]. In addition, a Singaporean study reported a higher prevalence among Indians compared to Chinese and Malays (30.7% versus 26.2%, and 25.5%, respectively) [23]. From the past studies, it could be argued that the association between ethnicity and the risk of DR is still inconclusive. The result may differ according to the geographical location of the patient and the lifestyle of the patient. Therefore, it is crucial to establish the association between local ethnicity and DR.

Aside from the sociodemographic information, DR is also commonly influenced by the patient’s comorbidities. Among the known comorbidities are hypertension, dyslipidaemia, overweight, and obesity [43,44]. An Indian study indicated that patients with hypertension are 2.14 times more likely to be diagnosed with DR than patients without hypertension [32]. Furthermore, the probability for a patient with dyslipidaemia to be diagnosed with DR is also higher due to the impact of lipids on the pathogenesis of DR [45,46]. Moreover, a clinical study reported that the severity of DR increased significantly with higher BMI (Grade 1: 26.50 ± 2.70, Grade 2: 28.11 ± 3.00, Grade 3: 28.69 ± 2.50) [47].

In managing DR patients, it is highly crucial to pay attention to the patient’s comorbidities. The risk of DR increases in the presence of other diabetic complications such as nephropathy, ischaemic heart disease, cerebrovascular disease, amputation, and diabetic foot ulcer [24,31,48,49]. In contrast, the likelihood of a patient with ischaemic heart disease being diagnosed with DR was nearly twice that of a patient without the condition [5]. In addition, the Wisconsin Epidemiological Study of DR (WESDR) found that proliferative DR was associated with stroke mortality in patients with type 1 and type 2 diabetes, as well as diabetes duration and glycaemic control [47,48,49,50,51]. In addition, findings from a past study reported that out of 100 patients with DFU, 90.0% of them were also diagnosed with DR [52]. The higher serum creatinine among DFU patients increased the risk of being diagnosed with DR [53]. Patients with a higher number of comorbidities are at a higher risk of being diagnosed with DR. Therefore, it is highly crucial for the patient to be assessed thoroughly during the clinical assessment for their diabetic management. However, in a lower- to middle-income country, the healthcare practitioner may not be able to access the patient’s medical record entirely due to the lack of data integration [54]. In addition, patients’ awareness of their comorbidities may also be insufficient, which will compromise the effectiveness of DM management and subsequently increase the risk of DR.

Lastly, DR is also linked to the diabetic condition of the patient. One of the important variables is the duration of DM diagnosis. A past study reported that a longer duration of diabetes might be related to a higher probability of the diagnosis of DR [14]. The United Kingdom Prospective Diabetic Study (UKPDS) study indicated that after six years of diagnosis, 22.0% of diabetic patients developed DR [29]. Furthermore, the HBA1c level also plays an important role in the diagnosis of DR. Participants with poor glycaemic control were found to be significant predictors of an increased risk of DR in a previous study (OR 3.37; 95% CI 2.13 to 5.34) [47].

In summation, it is apparent from the existing studies that the development of DR is influenced by four major factors, namely, sociodemographic characteristics, comorbidities, complications, and diabetic conditions. The effective management of the risk factors could reduce the prevalence of DR among type 2 DM patients. Therefore, in conducting future research related to DR, these four factors need to be studied carefully.

## 3. Research Gap

In Malaysia, 33.5% of diabetic patients are diagnosed with DR [25]. However, to date, there is a paucity of research that could guide healthcare practitioners in managing DR in Malaysia. Malaysia is a multiracial country with various types of geographical features, especially in the Borneo part of the country. The unique cultural background within each state may influence the people’s lifestyle and, subsequently, the management of DM. Therefore, specific research in a specific city is required to assess the incidence of DR and its risk factors.

Findings from the literature review have revealed insufficient scientific guidance in conducting a study on DR at the local level, in other lower- to middle-income countries, and in multiracial countries. Therefore, this study intended to close the gap by developing a conceptual framework for conducting DR-related studies that could be used in studies conducted in lower- to middle-income countries and among multiracial countries.

## 4. Suggested Conceptual Framework

Based on the literature review, the conceptual framework was developed as a scientific guide to conducting a study on DR (Figure 1). According to the literature review, four key factors were identified to increase the probability of being diagnosed with DR; sociodemographic characteristics, comorbidities, complications, and the diabetic condition of patient.

The first factor that was included in the proposed conceptual framework is the sociodemographic factor. To determine the association between sociodemographic factors and DR, this study suggests that the following three variables be explored thoroughly: age, gender, and ethnicity of the patients. The literature review has proven that patients in the older age group would be at a higher risk of developing DR. Furthermore, the probability of being diagnosed with DR is higher among men compared to women. In this study, these two variables would be investigated to determine if the same phenomena could be observed in Sabah or otherwise. In addition to these two variables, the ethnicity of the patient is also suggested to be analysed in detail. From the literature review, it is identified that most of the studies outside Southeast Asia do not consider ethnicity as a risk factor for DR. Nevertheless, studies from Southeast Asian countries such as Malaysia and Singapore have confirmed the association between ethnicity and the risk of DR, but the finding is still inconclusive due to the limited data. Therefore, it is crucial to include this variable in the proposed conceptual framework. This would subsequently institute structured scientific guidance for future research.

The second factor that was included in the proposed conceptual framework is the patient’s comorbidities. From the literature review, hypertension, dyslipidaemia, and obesity were proven by the existing literature to significantly influence the risk of DR. An in-depth examination of these three variables is critical because they correlate with the patient’s sociodemographic characteristics, particularly his or her ethnicity. A different ethnicity may practice different lifestyles and cultures that might influence the incidence of hypertension, dyslipidaemia, and obesity. This would subsequently impact the diagnosis of DR as well.

The proposed conceptual framework also suggested an in-depth analysis of the four types of diabetic complications. These complications are diabetic foot ulcer, nephropathy, cerebrovascular disease, and cardiovascular disease. Finally, the diabetic condition was also included in the proposed conceptual framework. The findings of the literature review have confirmed the impact of duration of disease and glycaemic condition on the diagnosis of DR. In the Malaysian setting, analysis of these variables is imperative as diabetic screening is commonly incomplete, especially in rural areas. In rural areas, there is also a lack of medical professionals, which may compromise the management of the disease for diabetic patients and subsequently influence the diagnosis of DR.

## 5. Proposed Study Framework According to the Proposed Conceptual Framework

In applying the proposed conceptual framework, this study is planning to conduct a cross-sectional study across diabetic patients residing in Sabah, Malaysia. To date, there is no scientific evidence on the impact of these complications on the diagnosis of DR involving Sabahans. In addition, according to the report on diabetic retinopathy produced by the Ministry of Health Malaysia, the prevalence of DR is the highest among Malay ethnicity, but the real situation concerning DR specifically in Sabah is still unknown. Sabah’s population consists of 33 indigenous groups, and each group carries its unique lifestyle and culture. In addition, the genetic predisposition also might be different from Peninsular Malaysia, as Sabah is an island situated in the South China Sea. The different ethnic proportions in Sabah compared to Peninsular Malaysia may draw a different finding from the state’s findings. In addition, diabetic screening in Sabah is also commonly incomplete compared to the other parts of Malaysia due to the difficulties in assessing the rural areas and the shortage of medical practitioners.

In the attempt to close the research gap and apply the proposed conceptual framework, a future study that involves the following primary objective will be conducted; to determine the prevalence of DR and identify the risk factors associated with DR among type 2 diabetes mellitus (DM) patients across primary health clinics in Sabah, Malaysia. In supporting the primary objective of the study, the specific objectives are developed as below:To identify the sociodemographic characteristics, comorbidities, and complications of T2DM patients in primary health clinics in Sabah;To identify the associated factors of DR among T2DM in Sabah primary health clinics;To determine the association between DR and sociodemographic data, such as age and gender;To determine the association between DR and comorbidities such as hypertension, dyslipidaemia, and high body mass index;To determine the association between DR and other diabetes complications such as nephropathy, foot ulcer, and cerebrovascular and cardiovascular disease;To determine the association between DR and above normal HBA1C level.

Secondary data will be retrieved from the Sabah Diabetes Registry between 2008 and 2015. Data stored in the Sabah Diabetes Registry consist of paediatric and adult data. Therefore, all type 2 diabetes mellitus (T2DM) patients, without regard to age preference or onset of diabetes, will be included in the proposed study. The focus on T2DM patients is due to the public health concerns about the high prevalence of T2DM patients in Malaysia, as it is listed as one of the major non-communicable diseases (NCDs) in Malaysia. In addition, type 1 diabetes mellitus (T1DM) is an autoimmune disease, which reduces its importance from the perspective of public health studies. All the findings from the proposed study will be proven statistically using bivariate and multivariate analysis. First, the relationship between diabetic retinopathy and risk factors will be performed by bivariate analysis of chi-square to see the associations between the outcome (diabetic retinopathy) and all the risk factors as illustrated in the proposed conceptual framework. Then, a multivariate logistic regression analysis will be applied to further confirm the findings. Figure 2 below summarises the proposed study.

The finding from the proposed study using the proposed conceptual framework is crucial to improving public health practice. The findings of the proposed study could become the guideline for policymakers to institute specific guidelines for DM management, especially among DR patients in Sabah. By identifying the risk factors, policymakers could adopt the most suitable technologies, such as continuous glucose monitoring (CGM) and therapies such as cytokine targeted therapy to improve the management of DM and, subsequently, DR in Sabah. In addition, this finding from the proposed study could also increase awareness of the importance of managing risk factors among diabetic patients.

## 6. Conclusions

The incidence of DR could be reduced by identifying the risk factors for this disease. It is critical that DR management be tailored to the patient’s specific needs. However, currently, there is a lack of scientific guidance to conduct scientific research on DR in Malaysia. From the literature review, this study has proposed a conceptual framework that could guide future research in conducting DR-related studies. In the proposed conceptual framework, four main risk factors were included, namely, sociodemographic characteristics, comorbidities, complications, and the diabetic condition of the patient. Each factor consists of different variables that require in-depth review and analysis. Currently, studies that explore the impact of ethnicity on the risk of DR are still lacking. Since the conceptual framework is developed by considering ethnicity as one of the important risk factors for DR, this conceptual framework would be beneficial for research conducted in lower- to middle-income countries and multiracial countries. In providing specific guidance to future research, this study has also proposed a study framework using the proposed conceptual framework. Although the proposed study framework is specific to a study in Sabah, the study framework is believed to be useful for future researchers to determine the prevalence of DR and its risk factors in other cities as well.

## Figures and Tables

**Figure 1 ijerph-19-12426-f001:**
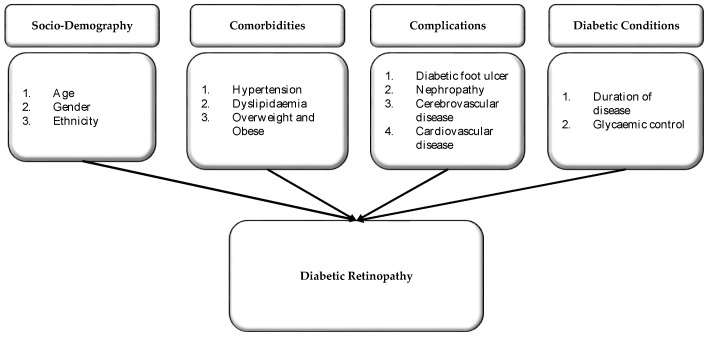
Suggested conceptual framework.

**Figure 2 ijerph-19-12426-f002:**
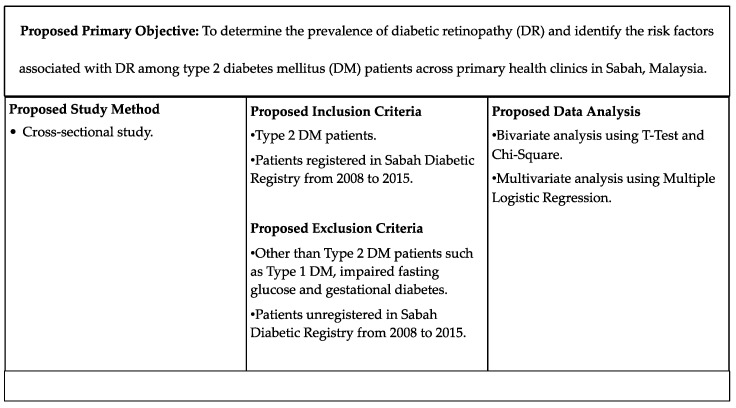
Suggested study to determine the prevalence of diabetic retinopathy (DR) and identify risk factors associated with DR among type 2 diabetes mellitus (DM) patients across primary health clinics in Sabah, Malaysia.

**Table 1 ijerph-19-12426-t001:** Inclusion and exclusion criteria.

Inclusion Criteria	Exclusion Criteria
Study was written in English language	Study was written in other than English language
Study conducted among type 2 diabetes patients	Study conducted among other than type 2 diabetes patients
Study conducted among diabetic retinopathy patients	Study on other diabetic complications

## Data Availability

Not applicable.

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
