# Peer review of "Study on the Development of a Conceptual Framework to Identify the Risk Factors of Diabetic Retinopathy among Diabetic Patients: A Concept Paper"

_ijerph, 2022, doi:10.3390/ijerph191912426_

Round 1

Reviewer 1 Report

 The authors tailored a conceptual framework to identify risk factors for diabetic patients in Sabah. The addressed topic is important. Especially sociodemographic factors are interesting as the data regarding this topic are limited. However I have some major concerns.

First of all, the study design is not well described. The authors plan to conduct cross-sectional study. It is not clear if the study group will consist of patients affected with type 1, type 2 diabetes or both. Does Sabah Diabetes Registry include only adults? Are there any exclusion criteria? Do researchers plan to use any specific statistical tests?

Besides, it would be interesting to describe the measures that the authors are planning to take after identifying the group of diabetic patients at high risk. Would it include only the treatment intensification? Will the authors encourage patients to use modern technologies like CGM to improve glycaemic control? Maybe some patients could receive cytokine-targeted therapy in the future like anti-VEGF treatment?

Moreover, I have some minor suggestions.

1.     Please explain the term “diabetic condition” in the abstract.

2.     The phrase “Malaysia is a multiracial country, especially in Borneo where there are various ethnicities” appears twice in the article. It is not necessary to repeat the sentence.

3.     The authors put the year of publication of several research papers in brackets. In my opinion it is not necessary.

4.     The paper “United Kingdom Prospective Diabetes Study” is not cited properly as the authors put the first names of researches before surnames.

5.     There is no reference number 27 in the text.

Author Response

Dear Reviewer 1.,

Thank you for your kind consideration to review our manuscript. We have carefully addressed all your valuable comments and suggestions. Please find attached the point-by-point response to your comments and suggestions. 

Regards.,

Athirah

Reviewer 2 Report

Since most of the references in this draft are not recent, but refer to information published a long time ago, it is difficult to share information on DR and analyze trends at present.

It is recommended to update most of the references to the latest.

Author Response

Dear Reviewer 2.,

Thank you for your kind consideration to review our manuscript. We have carefully addressed all your valuable comments and suggestions. Please find attached the point-by-point response to your comments and suggestions. 

Regards.,

Athirah

Reviewer 3 Report

To the authors to present the “Study on the Development of a Conceptual Framework to Identify the Risk Factors of Diabetic Retinopathy Among Diabetic Patients in Sabah: A Concept Paper”

Major concerns

#1This article lacks interest in the format presented, it does not provide anything new, it makes a review of the risk factors for diabetes in the introduction and its implications at the level of the retina, something quite well worked out in the existing literature on the subject.

#2 I do not understand that a study proposal is capable of publication, this paper should be part of the data extracted from the study that they suggest.

#3 They suggest studying a population between 2008 and 2015, is this patients who have been diagnosed with diabetes between those dates? They should be clear about the inclusion and exclusion criteria to carry out an investigation before starting, for example, duration of the disease or just the previous situation...

#4 The conceptual framework has neither results nor discussion.

# 5 Conclusions are based on an assumption.

Author Response

Dear Reviewer 3.,

Thank you for your kind consideration to review our manuscript. We have carefully addressed all your valuable comments and suggestions. Please find attached the point-by-point response to your comments and suggestions. 

Regards.,

Athirah

Reviewer 4 Report

The authors presented a concept paper with the future aim to identify the risk factors of Diabetic Retinopathy in Sabah. The idea of a concept paper may be interesting but the authors should extensively revise the paper and provide at the very beginning of the introduction detailed information about the notion of “concept paper”, indicating the specific reasons why the future study will be undertaken and motivations. In addition, specific details of the future study need to be added to the manuscript, in terms of timeline, and how it will be carried out. In details,  different paragraphs should be specifically added related to the future project scope, targets, timeline, milestones and management. Also, the authors should summarize in the introduction the different parts that will represent the sections of the paper so that the reader can have since the beginning more information about what the manuscript itself represents. Importantly, the entire manuscript should be revised for English style, grammar, punctuation. The language in general should be improved: the quality of the language is often poor and many terms and statements should be re-written in a more scientific and objective way. Also, additional information regarding the pathogenic mechanisms behind diabetes and its signs, symptoms, complications, should be provided. In summary, the manuscript should be extensively revised and re-written in many parts before publication can be considered. 

Author Response

Dear Reviewer 4.,

Thank you for your kind consideration to review our manuscript. We have carefully addressed all your valuable comments and suggestions. Please find attached the point-by-point response to your comments and suggestions. 

Regards.,

Athirah

Round 2

Reviewer 1 Report

The authors have thoroughly revised the manuscript. However the language style should be improved. I suggest to add the information why the patients with type 1 diabetes were excluded from the study. It is still unclear.

Author Response

Dear Reviewer 1,

Thank you for your positive and constructive comments. As suggested, we have improved the language style accordingly. Also, we have added justifications for the study sample. Please see page 5, Line 327 – 331. 

Thank you. 

Reviewer 2 Report

I approve the thesis because most needs have been met.

In particular, it is highly recommended to update the reference to the latest version.

Author Response

Dear Reviewer 2,

Thank you for your approval. We appreciate your constructive comments and feedback.

Thanks.

Reviewer 3 Report

This article needs to provide research data, , it does not provide anything new.

Author Response

Dear Reviewer 3,

Thank you for your feedback. The aim of this study is to produce a concept paper that would help future research in conducting research on Diabetic Retinopathy. In this concept paper, a conceptual framework has been developed that suits studies across Low- to middle-income countries. The suggested conceptual framework is the result of the study. 

Thanks.

Reviewer 4 Report

The authors did a great effort to revised the manuscript that has definitively improved compared to the original version. 

I would suggest to please revise once more the manuscript for English style and grammar and use of abbreviations (please check that the abbreviation is explained when first used in the manuscript and then only the abbreviation is used in the rest of the manuscript).

Author Response

Dear Reviewer 4,

Thank you for your constructive feedback and comments. As suggested, we have improved the language style accordingly. 

Thank you.